

# Transcriptome profiling reveals the role of ZBTB38 knock-down in human neuroblastoma

Jie Chen[1,2,3,*], Chaofeng Xing[1,2,*], Li Yan[4], Yabing Wang[5], Haosen Wang[6], Zongmeng Zhang[1], Daolun Yu[1], Jie Li[1], Honglin Li[7], Jun Li[1] and Yafei Cai[2]

[1] College of Life Sciences, Anhui Provincial Key Lab of the Conservation and Exploitation of Biological Resources, Anhui Normal University, WuHu, China
[2] College of Animal Science and Technology, Nanjing Agricultural University, Nanjing, China
[3] The Secondary Hospital of Wuhu, WuHu, China
[4] Department of Radiation Oncology, Linyi People Hospital, Linyi, China
[5] The First Affiliated Hospital of Wannan Medical College, WuHu, China
[6] Taizhou 4th Hospital, Taizhou, China
[7] Department of Biochemistry and Molecular Biology, Medical College of Georgia, Augusta State University, Augusta, GA, USA
* These authors contributed equally to this work.

Corresponding authors
Jun Li, lijunplant@163.com
Yafei Cai, ycai@njau.edu.cn

## ABSTRACT

ZBTB38 belongs to the zinc finger protein family and contains the typical BTB domains. As a transcription factor, ZBTB38 is involved in cell regulation, proliferation and apoptosis, whereas, functional deficiency of ZBTB38 induces the human neuroblastoma (NB) cell death potentially. To have some insight into the role of ZBTB38 in NB development, high throughput RNA sequencing was performed using the human NB cell line SH-SY5Y with the deletion of ZBTB38. In the present study, 2,438 differentially expressed genes (DEGs) in ZBTB38$^{-/-}$ SH-SY5Y cells were obtained, 83.5% of which was down-regulated. Functional annotation of the DEGs in the Kyoto Encyclopedia of Genes and Genomes database revealed that most of the identified genes were enriched in the neurotrophin TRK receptor signaling pathway, including PI3K/Akt and MAPK signaling pathway. we also observed that ZBTB38 affects expression of CDK4/6, Cyclin E, MDM2, ATM, ATR, PTEN, Gadd45, and PIGs in the p53 signaling pathway. In addition, ZBTB38 knockdown significantly suppresses the expression of autophagy-related key genes including PIK3C2A and RB1CC1. The present meeting provides evidence to molecular mechanism of ZBTB38 modulating NB development and targeted anti-tumor therapies.

## INTRODUCTION

Neuroblastoma (NB) is an embryonal malignant tumor that originates from the neural crest cells of the sympathetic nervous system. NB is the most common extracranial solid tumor in children, accounting for 8–10% of pediatric malignancies (*Castel, Segura & Berlanga, 2013*; *Schulte et al., 2013*). This disease is highly malignant

and progresses rapidly—most of the patients would have been at the advanced stage upon diagnosis when conventional radiotherapy and chemotherapy would feature low efficacy, thus resulting in extremely low survival rate (*Bagatell & Cohn, 2016*). In recent years, new therapeutic methods, including hematopoietic stem cell transplantation and biological immunotherapy, have been employed to treat relapsed or refractory NB, but their efficacy remains limited (*Binkhathlan & Lavasanifar, 2013*; *Han & Wang, 2015*). Drug resistance has been recognized as the key obstacle to reaching a satisfactory outcomes (*Castel, Segura & Berlanga, 2013*), whereas the induction of programmed death of the cancer cell death by targeted gene therapy shows notable potential in improving the cure rate and long-term survival of NB patients, especially those with higher risks.

ZBTB38 belongs to the zinc finger and BTB domain-containing protein family (Pfam). Most members of the family, as transcription factors, bind to specific DNA sequences, and regulate the transcriptional activity of target genes (*Sasai et al., 2005*; *Stogios et al., 2005*). The ZBTB family members also participate in various intracellular signal transduction pathways via recognizing and interacting with other proteins, thereby playing important roles in the transcriptional repression, DNA damage, tumorigenesis, cell proliferation, differentiation, and apoptosis (*Lee et al., 2010*; *Oikawa et al., 2008*; *Nishii et al., 2012*). At least 49 ZBTB proteins are encoded in the human genome; most of them are nuclear proteins (*Lee & Maeda, 2012*). Among the predicted BTB domain-containing proteins encoded by the human genome, only several of them have been functionally characterized (*Cai et al., 2012*; *Oikawa et al., 2008*). No relevant studies have been reported concerning the effect of ZBTB38 on human NB.

Transcriptomic studies have progressed rapidly over the recent years, which, contrary to the research on an individual genes, enabled the investigation on the altered expression of differentially expressed genes (DEGs) at the level of whole protein-coding or non-coding RNAs in cells or tissues of the body. Transcriptomic studies can also provide information on the relationship between the transcriptional regulation and the protein functions in the whole genome under specific conditions (*Reimann et al., 2014*; *Zhao et al., 2011*). Next-generation sequencing (NGS) technology offers important technical support for the annotation and quantification of transcriptomes. The major strength of this technique lies in its high throughput and high sensitivity for transcript abundance, thus providing thorough understanding of the transcriptional genome information in a comprehensive manner and valuable resources for the investigation of the therapeutic biomarkers of cancer (*Chang et al., 2015*; *Li et al., 2014*).

Therefore, in the present study, a high-throughput transcriptome sequencing approach was adopted to investigate the transcriptome profiles of NB cells in which the expression of ZBTB38 gene was down-regulated, thus revealing the potential biomarkers associated with anti-tumor therapies for NB.

# MATERIALS AND METHODS

## Cell culture and standard assays

SH-SY5Y cells were purchased from American Type Culture Collection (Rockville, MD, USA) and cultured in Dulbecco's modified Eagle's medium supplemented with 10% fetal

bovine serum and penicillin–streptomycin. siRNAs were used to deplete ZBTB38 gene in SH-SY5Y cells by lipofection. Scramble siRNA were used as negative control for knockdown experiment. Transient transfections, quantitative real-time polymerase chain reaction (qRT-PCR) and western blotting were performed as described previously (Cai et al., 2012, 2017). GAPDH and β-actin (for human samples) were used as the internal control. The primers used in qRT-PCR and siRNA suppression assays are listed in Table S1.

Cell viability was tested using the CCK-8 assay. The absorbance of each well was measured at 450 nm on a microplate reader. The proliferation rate was defined in terms of the percentage of each group of surviving cells compared with the untreated group for both cell lines.

The Cell Death Detection Elisa Kit (Roche, Indianapolis, IN, USA) was used to determine apoptosis by measuring mono and oligonucleosomes in the lysates of apoptotic cells according to the manufacturer's protocol.

ZBTB38 were purchased from NovusBio (Littleton, CO, USA). PTEN, GAPDH, β-actin RAPTOR, LC3B, p62, and RB1CC1 antibodies were procured from Abcam plc (Boston, MA, USA). In Situ Cell Death Detection Kit-POD was purchased from Roche (Basel, Switzerland).

## RNA preparation and library construction for transcriptome sequencing

Transcriptome high-throughput sequencing was performed in the control group (SH-SY5Y cells transfected with liposome alone, Samples-ID: T04, T05, T06) and the treatment group (SH-SY5Y cells transfected with ZBTB38 siRNA, Samples-ID: T01, T02, T07). Total RNA was isolated from SH-SY5Y cells using TRIzol and the pure-link RNA mini kit (ThermoFisher Scientific, Waltham, MA, USA) according to manufacturer's instructions. RNA purity was checked using the NanoPhotometer spectrophotometer (Implen, Inc., Westlake Village, CA, USA). RNA concentration was measured using the Qubit RNA Assay Kit in Qubit 2.0 Fluorometer (Life Technologies, Camarillo, CA, USA). RNA integrity was assessed using the RNA Nano 6000 Assay Kit of the Agilent Bioanalyzer 2100 system (Agilent Technologies, Santa Clara, CA, USA).

In total, two μg RNA per sample was used as input material for RNA sample preparations. This study included two groups of three biological replicates. Sequencing libraries were generated using a NEBNext UltraTM RNA Library Prep Kit for Illumina (NEB, Beverly, MA, USA), and index codes were added to attribute sequences to each sample. Fragmentation was performed using divalent cations under elevated temperature in NEBNext First Strand Synthesis Reaction Buffer ($5\times$). First-strand cDNA was synthesized using random hexamer primer and M-MuLV Reverse Transcriptase (RNase H). Second-strand cDNA synthesis was subsequently performed using DNA Polymerase I and RNase H. Remaining overhangs were converted into blunt ends via exonuclease/polymerase activities. After the adenylation of 3′ ends of DNA fragments, NEBNext Adaptor with a hairpin loop structure was ligated to prepare for hybridization. The library fragments were purified using AMPure XP System (Beckman Coulter, Beverly, MA, USA).

Then, three μl USER Enzyme (NEB, USA) was used with size-selected, adaptor-ligated cDNA at 37 °C for 15 min, followed by 5 min at 95 °C before PCR. Following this, PCR was performed with Phusion High-Fidelity DNA polymerase, universal PCR primers, and index (X) Primer. Finally, PCR products were purified (AMPure XP System, Beckman Coulter, Beverly, MA, USA), and library quality was assessed using the Agilent Bioanalyzer 2100 system. The clustering of the index-coded samples was performed on a cBot Cluster Generation System using the TruSeq PE Cluster Kit v4-cBot-HS (Illumia) (NEB, Beverly, MA, USA). Following cluster generation, the library preparations were sequenced on an Illumina Hiseq 2500 platform, and paired-end reads were generated.

## Data and statistical analysis

### Cancer genomics analysis

We downloaded the mRNA expression data from the cancer genome atlas (TCGA) database, and systematically evaluated the expression of ZBTB38 and correlation with patients' survival in tumors of the TCGA database.

### Quality control

Raw reads of fastq format were firstly processed through in-house perl scripts. In this step, clean reads were obtained by removing reads containing adapter, reads containing ploy-*N* and low-quality reads from raw reads. At the same time, Q20, Q30, GC-content and sequence duplication level of the clean reads were calculated. All the downstream analyses were based on clean reads with high quality (*Ewing & Green, 1998*; *Ewing et al., 1998*). The clean data of this article are publicly available in the NCBI sequence reads archive with accession number SRP150042.

### Comparative analysis

The adaptor sequences and low-quality sequence reads were removed from the data sets. Raw sequences were transformed into clean reads after data processing. These clean reads were then mapped to the reference genome sequence. Only reads with a perfect match or one mismatch were further analyzed and annotated based on the reference genome. Tophat2 tools were used to map with reference genome (*Kim et al., 2013*; *Langmead et al., 2009*). Reference genome download address: ftp://ftp.ensembl.org/pub/release-80/fasta/homo_sapiens/.

### Gene functional annotation

The assembled sequences were compared against the NCBI non-redundant protein sequences (NR), Pfam, clusters of orthologous groups of proteins (KOG/COG), a manually annotated and reviewed protein sequence database (Swiss-Prot), KEGG ortholog database (KO), and gene ontology (GO) databases with an $E$-value $\leq 10^{-5}$ for the functional annotation. The Blast2GO program was used to obtain GO annotation of unigenes including molecular function (MF), biological process (BP), and cellular component (CC) categories (*Gotz et al., 2008*).
### Differential expression analysis

Differential expression analysis of the two conditions was performed using the DEGseq R package (*Robinson, McCarthy & Smyth, 2010*). The *p*-values obtained from a negative binomial model of gene expression were adjusted using Benjamini and Hochberg corrections to control for false discovery rates (*Anders & Huber, 2010*). Genes with an adjusted *p*-value < 0.05 were considered to be differently expressed between groups. DEG expression levels were estimated by fragments per kilobase of transcript per million fragments mapped (*Florea, Song & Salzberg, 2013*). The formula is shown as follow:

$$FPKM = \frac{cDNA\ fragments}{Mapped\ fragments\ (millions) \times Transcript\ length\ (kb)}$$

### GO enrichment and KEGG pathway enrichment analysis

Gene ontology enrichment analysis of the DEGs was implemented in the "GOseq" package in R based on a Wallenius non-central hyper-geometric distribution, which can adjust for gene length bias in DEGs (*Young et al., 2010*).

KEGG is a database for understanding high-level functions and utilities of biological systems through large-scale molecular datasets generated by genome sequencing and other high-throughput experimental technologies (http://www.genome.jp/kegg/) (*Kanehisa et al., 2008*). We used the KOBAS software to test for the statistical enrichment of DEGs in KEGG pathways. KEGG enrichment can identify the principal metabolic pathways and signal transduction pathways of DEGs (*Mao et al., 2005*).

### DEGs quantitative real-time PCR verification

For validation of the transcriptome result, we subjected three significantly differential expressed unigenes on related pathways to qRT-PCR analysis. Redundant RNA from the cDNA library preparation was used to perform reverse transcription according to the Invitrogen protocol. qRT-PCR were performed as described previously (*Zhang et al., 2017*). The primers used in qRT-PCR suppression assays are listed in Table S1.

### Statistical analysis

All data were reported as mean ± standard deviation and analyzed using one-way analysis of variance in SPSS v.17.0. Statistical tests were performed with the Kruskal–Wallis and Mann–Whitney *U*-tests. A least significant difference test was used for comparisons between groups. A *p*-value < 0.05 was considered statistically significant.

## RESULTS

### Variation of ZBTB38 in tumors

According to the statistical analysis of the TCGA database resources, we found that the expression changes of ZBTB38 gene are closely related to the occurrence of 20 kinds of cancers, and especially the most remarkable down-regulated expression in uterine corpus endometrial carcinoma (UCEC) and cervical squamous cell carcinoma and endocervical adenocarcinoma (CESC) (*p* < 0.05) (Fig. 1). However, in the prognosis of

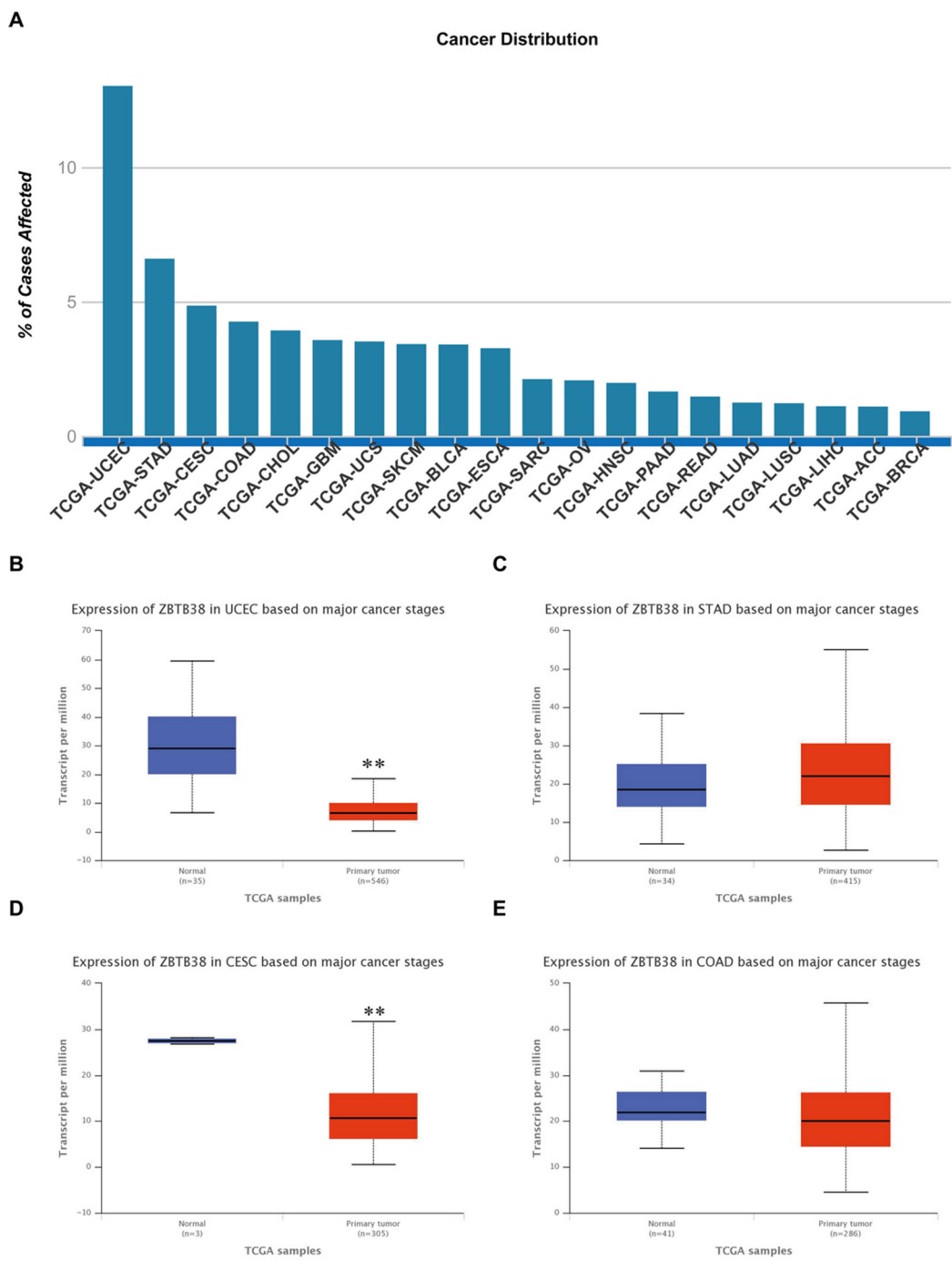

**Figure 1 Expression analysis of ZBTB38 gene in different tumors based on TCGA database.** (A) The expression changes of ZBTB38 gene are closely related to the occurrence of 20 kinds of cancers; (B–E) ZBTB38 expression profiles based on top four cancer stages. $^{**}p < 0.01$.

these 20 tumors, we only uncover here that low expression of ZBTB38 was associated with improved the prognosis of the brain lower grade glioma (LGG) patients (Fig. 2), suggesting that these changes are closely related to neuronal tumors.

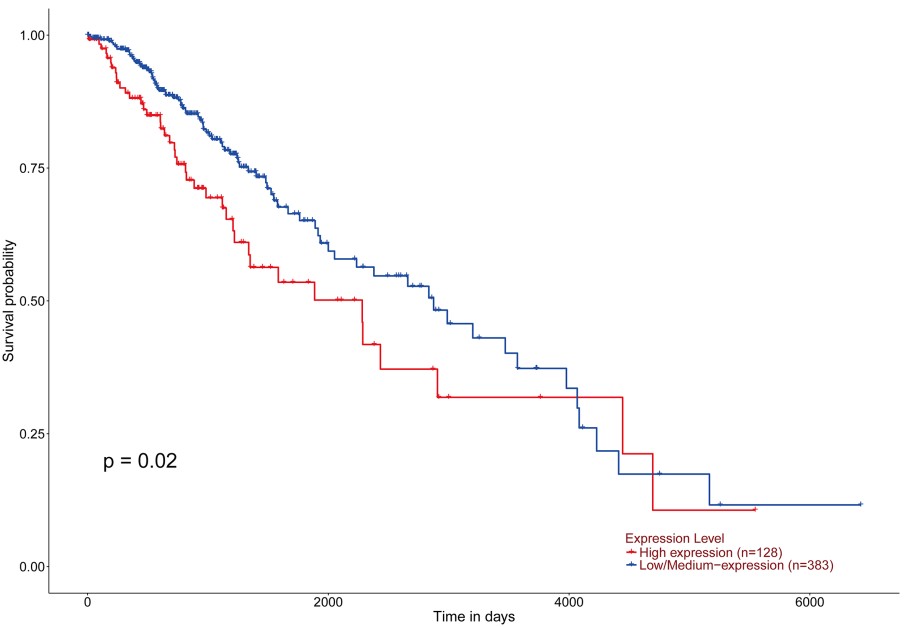

**Figure 2** **Effect of ZBTB38 expression level on LGG patient survival.** Red and blue lines indicated high and low expression groups, respectively. $p = 0.02 < 0.05$ was considered to be statistically significant.

## Neuroblastoma cell proliferation and viability after down-regulated of ZBTB38 expression

To investigate the importance of ZBTB38 in the process of neuronal tumors, three pairs of siRNAs named siRNA1, siRNA2, and siRNA3, were designed to suppress expression of ZBTB38 in human NB cells SH-SY5Y. The protein level of ZBTB38 was decreased significantly ($p < 0.05$) at 24 h after transfection (Figs. 3A and 3B), and furthermore, siRNA3 worked best for the suppression. No significant difference in cell proliferation and viability was observed among the initial phases of each group after culture by transient transfection ($p > 0.05$). From 12–72 h, the ZBTB38$^{-/-}$ SH-SY5Y group showed significantly lower cell proliferation and viability than the control group ($p < 0.05$) (Figs. 3C and 3D). Whereas, a sharp increase in apoptosis of SH-SY5Y cells were observed following ZBTB38 siRNA exposure (Fig. 4A). We next determined the expression levels of pro-apoptotic genes in ZBTB38 knockdown cells compared with knockdown control cells. The knockdown of the ZBTB38 gene resulted in a decrease in the expression levels of Noxa, Bak, Bim, Puma, and DR5 genes, with significant differences in Noxa, Bim, and DR5 ($p < 0.05$) (Fig. 4B). These data indicate that inhibition of ZBTB38 triggers apoptosis of NB cells.

## Quality control and yield statistics of transcriptome sequencing data

A total of 47.05 Gb clean data were obtained through the transcriptome sequencing of SH-SY5Y cells, with at least 6.12 Gb and $a \geq 89.30\%$ Q30 percentage for each sample (Table 1). Efficiency of sequence alignment referred to the percentage of mapped reads in the clean reads, which reflected the utilization of transcriptome sequencing data.

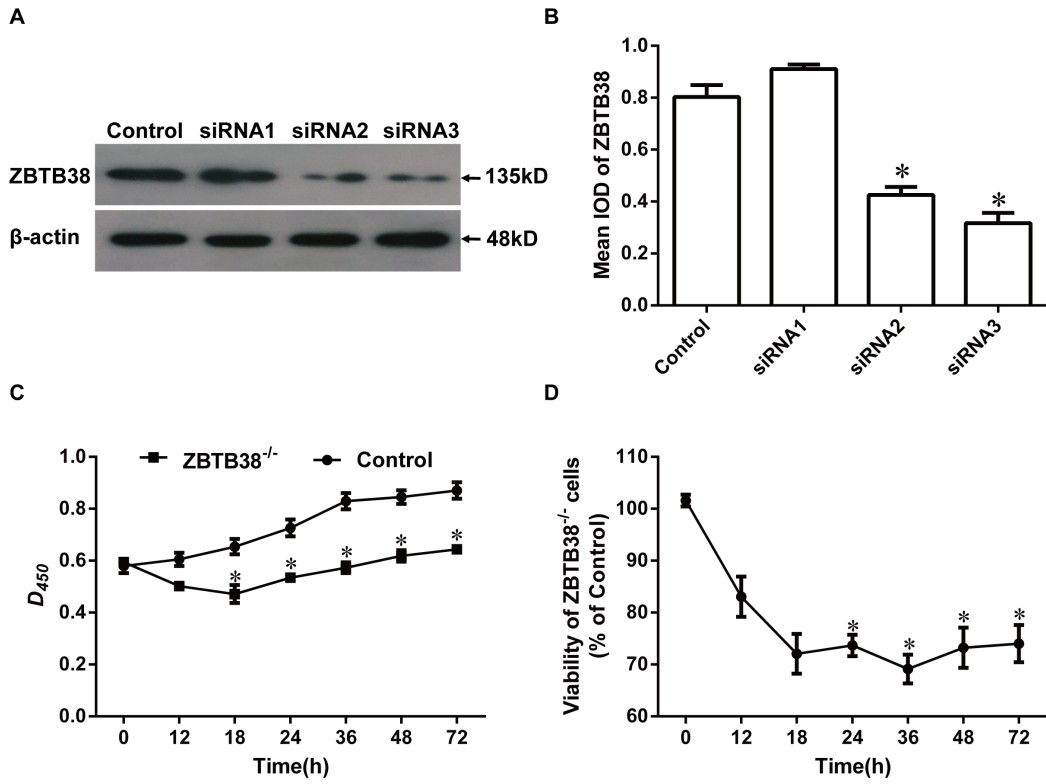

**Figure 3 Proliferation and viability of ZBTB38 knockdown SH-SY5Y cells.** (A and B) SH-SY5Y cells were divided into two groups underwent control group and siRNA, respectively. Cell lysates were collected for Western blot analysis on the 72 h after transfecting different sequences of ZBTB38 siRNAs (siRNA1, siRNA2, and siRNA3 indicate three different siRNA primers). Representative images of these assays are shown in (A) and quantitative data are shown in (B); β-actin was used as an internal control. SH-SY5Y cell proliferation (C) and viability (D) in different groups. *$p < 0.05$. Data are presented as means ± SEM from at least three independent experiments. Control, SH-SY5Y cells treated with liposome alone; ZBTB38$^{-/-}$, SH-SY5Y cells transfected with ZBTB38 siRNA.

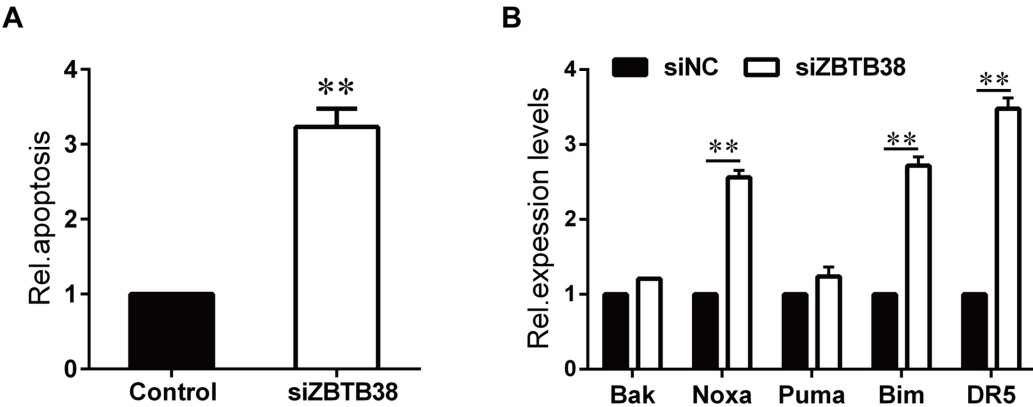

**Figure 4 Loss of ZBTB38 induces apoptosis in SH-SY5Y cells.** (A) Apoptosis of SH-SY5Y cells in the presence or absence of ZBTB38 siRNA was determined by The Cell Death Detection Elisa Kit. (B) RNA from ZBTB38 knockdown SH-SY5Y cells (siZBTB38) and control cells (siNC) was collected for QRT-PCR analysis to determine the expression levels of pro-apoptotic genes. **$p < 0.01$.

**Table 1 Summary of sequence comparisons among sample sequencing data and selected reference genomes.**

| Samples-ID | Total reads | Mapped reads | Uniq mapped reads | Multiple map reads | GC content | % ≥ Q30 |
|---|---|---|---|---|---|---|
| T01 | 41,320,306 | 32,993,483 (79.85%) | 29,171,906 (70.60%) | 3,821,577 (9.25%) | 56.24% | 89.44% |
| T02 | 53,706,092 | 42,655,511 (79.42%) | 37,610,672 (70.03%) | 5,044,839 (9.39%) | 55.36% | 89.45% |
| T04 | 54,889,928 | 44,964,104 (81.92%) | 41,561,590 (75.72%) | 3,402,514 (6.20%) | 52.17% | 90.04% |
| T05 | 50,160,236 | 40,559,313 (80.86%) | 37,552,835 (74.87%) | 3,006,478 (5.99%) | 52.16% | 90.25% |
| T06 | 62,721,676 | 50,526,963 (80.56%) | 47,347,475 (75.49%) | 3,179,488 (5.07%) | 51.94% | 90.05% |
| T07 | 54,693,534 | 43,544,178 (79.61%) | 38,543,679 (70.47%) | 5,000,499 (9.14%) | 55.22% | 89.30% |

Note:

T01, T02, and T07 indicate the ZBTB38$^{-/-}$ SH-SY5Y cells. T04, T05, and T06 indicate the control groups. Total reads: number of clean reads, single-ended; mapped reads: number of reads aligned to the reference genome and percentage in clean reads; uniq mapped reads: match the number of reads to the unique position of the reference genome and the percentage of the clean reads; multiple map reads: the number of reads aligned to multiple locations in the reference genome and the percentage of clean reads. GC content: the clean data GC content; ≥ Q30%: the percentage of bases with a clean data quality value ≥30.

Statistical analysis of the alignment results showed that the efficiency of read alignment for the reads of each sample and the reference genome ranged between 79.42% and 81.92% (Table 1), which guaranteed that the selected reference genome assembly was qualified for data analysis.

Qualified transcriptome libraries are a major requisite for transcriptome sequencing. To ensure the quality of the libraries, quality of the transcriptome sequencing libraries was evaluated from three different perspectives:

(1) Randomicity of mRNA fragmentation and the degradation of mRNA were evaluated by examining the distribution of inserted fragments in genes. As shown in Fig. S1, the degradation of mRNAs was relatively low in the six groups of samples.

(2) The dispersion degree of the inserted fragment length directly reflected the efficiency of magnetic bead purification during library preparation. Simulated distribution of the inserted fragment length for each sample showed only single-peak pattern, indicating a high purification rate (Fig. S2).

(3) With the increase of sequencing data, the number of DEGs tended to saturate, as shown in Fig. S3, which confirmed that the data were sufficient and qualified for the subsequent analysis.

### DEG and DEGs function annotation

To acquire the comprehensive genetic information of ZBTB38$^{-/-}$ SH-SY5Y cells, the unigenes were blasted against the NR, Swiss-Prot, GO, COG, KOG, Pfam, KEGG database resources to identity the functions of all of the unigene sequences. All of DEGs were annotated to genes having known functions in the indicated databases based on the sequences with the greatest similarity. DEseq was used to analyze the DEGs derived from the two groups of cells to obtain a DEGs set. Finally, a total of 2,036 (83.5%) down-regulated DEGs and 402 (16.5%) up-regulated DEGs were selected (Table S2; Fig. S4). The number of DEGs annotated in this gene set was shown in Table 2.

A total of 2,258 (93.4%) DEGs were annotated successfully by GO annotation. These annotated DEGs were classified into the next terms of three ontologies: BP, CC, and MF. The distribution of unigenes is shown in Fig. 5. Among the "BP," a high

**Table 2 Summary of the function annotation results for ZBTB38$^{-/-}$ unigenes in public protein databases.**

| DEG set | Total | COG | GO | KEGG | KOG | NR | Swiss-Prot | eggNOG |
|---------|-------|-----|-----|------|-----|-----|-----------|--------|
| T04_T05_T06 vs T01_T02_T07 | 2,417 | 999 | 2,258 | 1,512 | 1,733 | 2,337 | 2,377 | 2,405 |

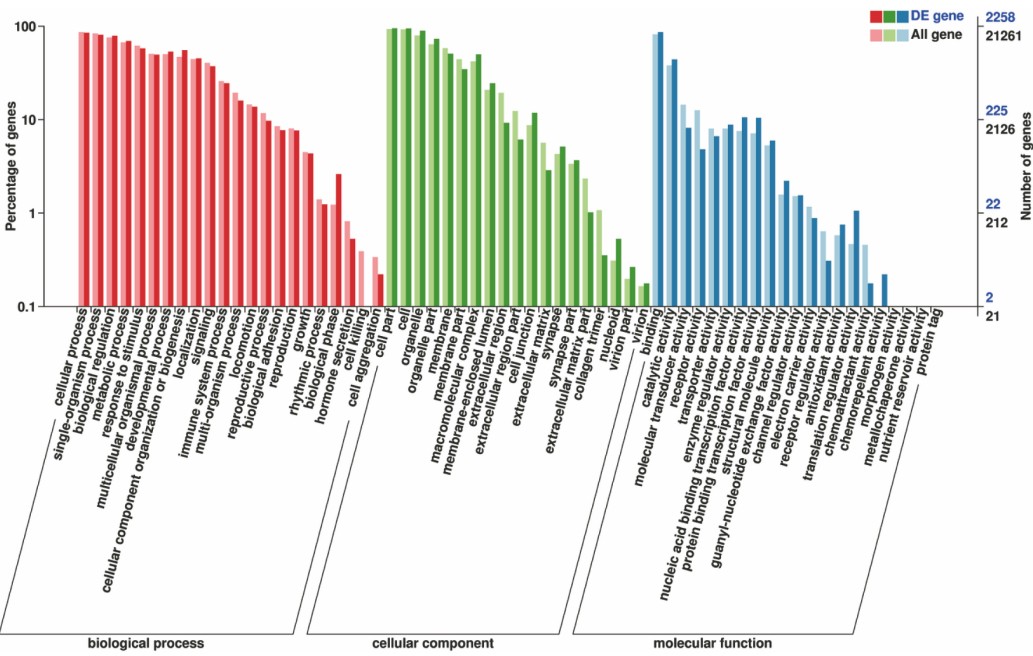

**Figure 5 Gene function classification of all annotated unigenes by gene ontology.** The vertical axis represents the number of unigenes, and horizontal axis gives the specific GO sub-categories.

percentage of genes were classified into cellular process (1,924 unigenes, 85.2%). Within the CC category, the majority of genes were assigned into cell part (2,145 unigenes, 95%). For the MF, most of genes were involved in "Binding" (1,949 unigenes, 86.3%).

The greatest number of annotated unigenes were involved in BP. The results of the topGO functional enrichment analyses of DEGs indicated that the most significantly enriched GO terms focus on "neurotrophin TRK receptor signaling pathway" (Table 3).

The unigenes was blasted against the COG database in order to orthologously classify gene products. COG classification statistical results of DEGs were shown in Fig. 6. In addition to "general function prediction only," "replication, recombination, and repair" accounted for the largest proportion of unigenes (180 DEGs, 13.06%), followed by "transcription" (133 DEGs, 9.65%), "signal transduction mechanisms" (128 DEGs, 9.29%), "translation, ribosomal structure, and biogenesis" (80 DEGs, 5.81%), "posttranslational modification, protein turnover, and chaperones" (79 DEGs, 5.73%), "cell cycle control, cell division, and chromosome partitioning" (44 DEGs, 2.98%). According to the annotation results of the DEGs KEGG database, the largest proportion of the unigenes were involved in the "MAPK signaling pathway" and "PI3K-Akt signaling pathway" of "environmental information processing" (Fig. 7).

**Table 3 TopGO enrichment results of differential expression genes.**

| GO.ID | Term | Annotated | Significant | Expected | KS |
|---|---|---|---|---|---|
| GO:0048011 | Neurotrophin TRK receptor signaling pathway | 562 | 68 | 60.42 | 9.00E-14 |
| GO:0045893 | Positive regulation of transcription, DNA-templated | 2,389 | 314 | 256.83 | 2.90E-13 |
| GO:0045944 | Positive regulation of transcription from RNA polymerase II promoter | 1,654 | 212 | 177.81 | 4.60E-13 |
| GO:0007268 | Synaptic transmission | 1,693 | 160 | 182 | 6.90E-13 |
| GO:0044281 | Small molecule metabolic process | 5,268 | 511 | 566.33 | 2.20E-11 |
| GO:0046777 | Protein autophosphorylation | 510 | 75 | 54.83 | 2.80E-11 |
| GO:0007173 | Epidermal growth factor receptor signaling pathway | 597 | 79 | 64.18 | 5.40E-11 |
| GO:0051656 | Establishment of organelle localization | 626 | 123 | 67.3 | 1.90E-10 |
| GO:0019219 | Regulation of nucleobase-containing compound metabolic process | 7,440 | 1,046 | 799.83 | 4.40E-10 |

**Note:**
Term: GO function; annotated: the number of genes annotated to this function for all genes; significant: the number of genes annotated to this function in the DEG; expected: the expected value of the number of DEGs for this function; KS: statistical significance of enriched term, the smaller the KS value, the more significant the enrichment.

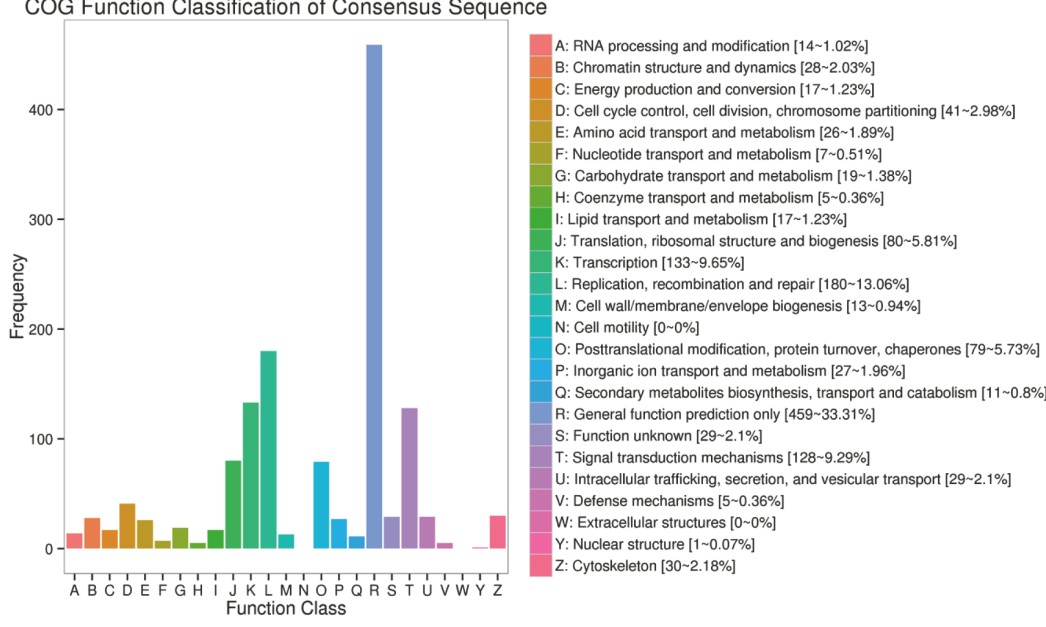

**Figure 6 COG function classification of consensus sequence.** The vertical axis represents the frequency of unigenes classified into the specific categories, and horizontal axis gives the COG function classification.

Based on the results above, a large number of DEGs were screened after a comparative analysis of relevant databases. Meanwhile, functional annotation was also carried out that was crucial for the further understanding of the cellular functions of ZBTB38 gene as a transcription factor.

## Detection of candidate genes and analysis of the results of real-time quantitative PCR

We analyzed whether the DEGs were over-presented on a pathway by enrichment of DEGs KEGG pathway (Fig. S5), taking FPKM as a measure for the level of transcripts or gene

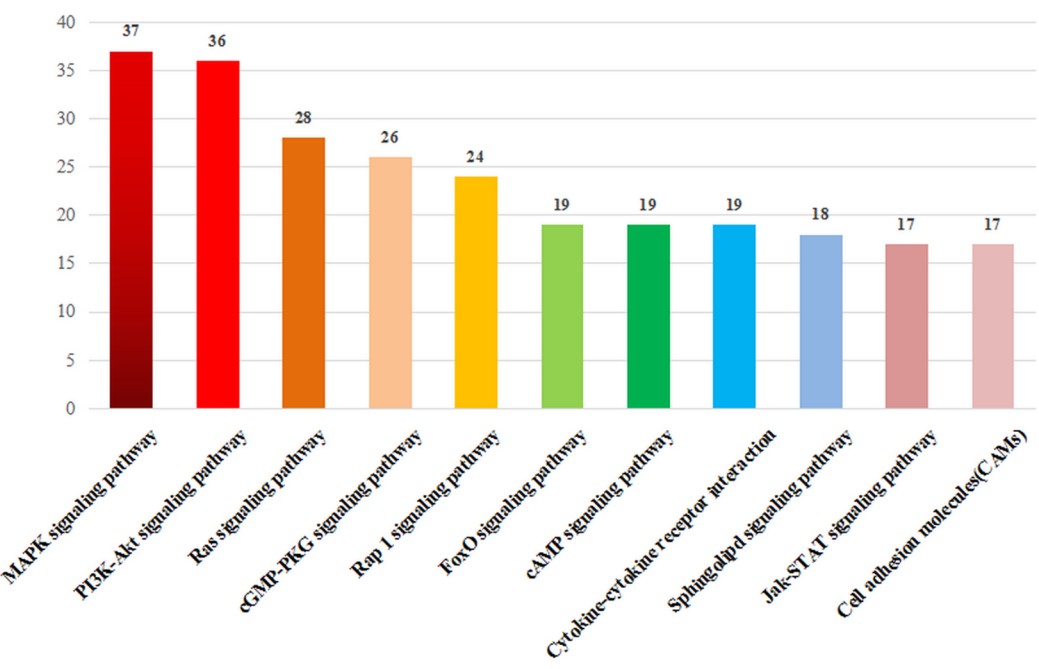

**Figure 7 DEG KEGG classification.** The vertical axis lists the various metabolic pathways, and horizontal axis gives the number of annotated genes in the pathways.

expressions, DEGs in the p53 signaling pathway, including CDK4/6 (ENSG00000105810), Cyclin E (ENSG00000175305), MDM2 (ENSG00000135679), ATM (ENSG00000149311), ATR (ENSG00000175054), PTEN (ENSG00000171862), were down-regulated, and Gadd45 (ENSG00000179271) and PIGs (ENSG00000115129) were up-regulated (Table S2; Fig. S5).

Top 20 down-regulated unigenes associated with autophagy were selected (Table S3), among which PIK3C2A was the most down-regulated one, followed by RB1CC1 gene. In summary, the transcription factor ZBTB38 is involved in the process of protein synthesis and also, as a positive regulatory factor, in the occurrence of autophagy directly.

To validate the sequencing results obtained by RNA-seq, real-time quantitative PCR was performed on three candidate genes, including PIK3C2A, RB1CC1, ATM, related to the mTOR signaling pathway. The result showed that the expression of these candidate genes was significantly decreased in the ZBTB38$^{-/-}$ cells compared to control group, which was similar to the RNA-seq data (Fig. 8). The result verified the reliability of the transcription sequencing results.

To further explore the mechanism involved in these events, we examined the expression levels of autophagy genes, in the presence or absence of ZBTB38. The expression levels of LC3B and RB1CC1 were significantly decreased in human NB cells after ZBTB38 knockdown, compared with those from empty liposome-treated SH-SY5Y cells (Fig. 9), which indicate that autophagy is inhibited. In addition, we also detected PTEN and RAPTOR, which are key genes in mTORC1 regulation of autophagy signaling pathway, their expression levels were decreased in ZBTB38 siRNA-treated SH-SY5Y cells, whereas

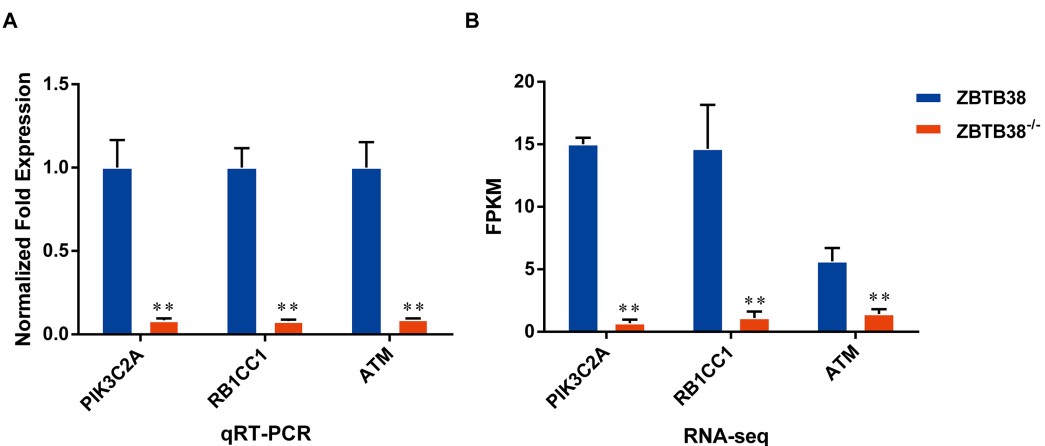

**Figure 8 Differential expression analysis of candidate genes between ZBTB38$^{-/-}$ and ZBTB38 SH-SY5Y cells.** (A) The result of qRT-PCR. (B) The result of RNA-seq. $^{**}p < 0.01$.

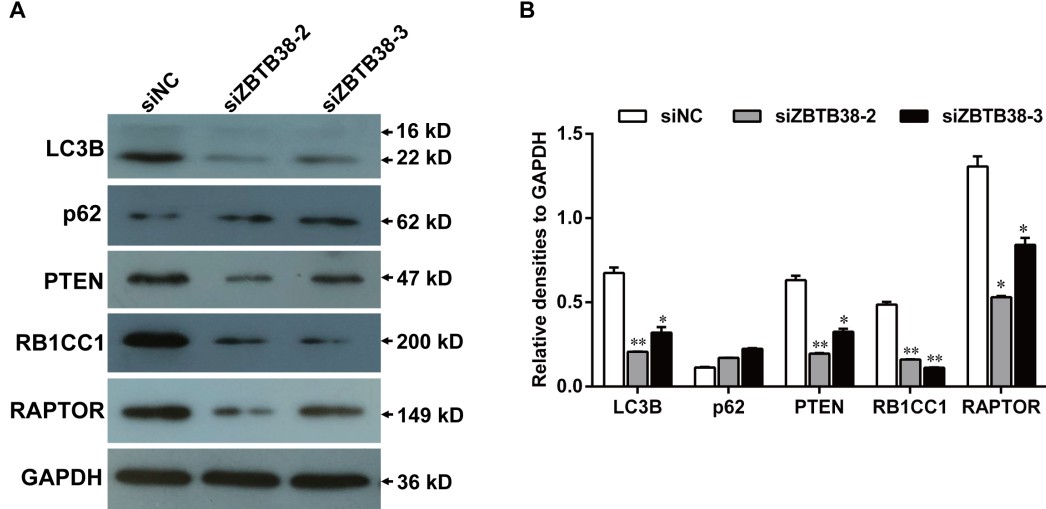

**Figure 9 Knockdown of ZBTB38 triggers inhibition of autophagy in SH-SY5Y cells.** (A) SH-SY5Y cells were transfected with scramble siRNA (siNC) or siRNA against ZBTB38 (siZBTB38-2 and siZBTB38-3) for 72 h and cell lysates were collected for Western blot analysis. (B) Quantification of protein expression was performed by densitometric analysis. $^{*}p < 0.05$; $^{**}p < 0.01$.

p62 expression was increased. These results indicate that ZBTB38 loss-mediated autophagy inhibition is likely associated with activation of the mTORC1 signaling pathway. In summary, the transcription factor ZBTB38 is involved in the process of protein synthesis and also, as a positive regulatory factor, in the occurrence of autophagy directly.

## DISCUSSION

Transcriptomic studies have progressed rapidly in recent years. Based on the information of the whole mRNAs obtained in one cell or tissue, the transcriptomic studies provide data on the expression regulation systems and protein functions of all genes.

NGS facilitates the deep sequencing of whole cancer genomes for the discovery of novel therapeutic biomarkers, helping to consequently build a solid foundation for the comprehensive studies of cancer pharmacogenetics. Furthermore, NGS allows for detailed analyses of the whole epigenome and transcriptome, thus profoundly revealing the multilevel regulation networks of the human genome (*McGettigan, 2013*; *Wang, Gerstein & Snyder, 2009*; *Young et al., 2010*). Remarkably, the large amount of data on gene expression profiles revealed by transcriptome sequences have provided valuable resources for studies investigating the therapeutic biomarkers of cancer.

The genomic instability is an important factor in the early stages of cell carcinogenesis. This property primarily results from the error-prone DNA repair and the accumulation of abnormal DNA after repair are the main causes of genomic instability that follows (*Lord & Ashworth, 2012*). The strict regulation of gene transcription is an essential factor to maintain the genomic stability (*Kakarougkas et al., 2014*). According to the statistical analysis of the TCGA database, the ZBTB Pfam is mainly involved in the expression regulation of the target genes. The amplification, deficiency and/or mutation of most genes in the ZBTB family occurs in different types of tumors (*Jardin et al., 2007*; *Maeda et al., 2007*; *Phan & Dalla-Favera, 2004*). Among ZBTB genes, the expression changes of ZBTB38 gene are closely related to the occurrence of 20 kinds of cancers (Fig. 1A), and different tumors exhibit significant differential expression changes, especially the remarkably down-regulated expressions of UCEC and CESC (Figs. 1B and 1D). However, in our study, the statistical analysis of the prognosis of the LGG patients exhibited a negative correlation with the expression changes of ZBTB38 (Fig. 2), indicating a significant concern regarding the study of the effects of ZBTB38 expression changes on the occurrence and development of neuroma. However, there was no relevant report focused on the expression change of ZBTB38 in NBs. This study demonstrated for the first time that the in vitro knockdown of ZBTB38 seriously affected the proliferation of NB cells. Accordingly, the biological function of ZBTB38 and its relationship with the clinical prognosis of NB deserves further analysis.

The annotation of the DEGs function revealed that after ZBTB38 knockdown, the most of DEGs were enriched in the neurotrophin TRK receptor signaling pathway. Neurotrophic factors (NTs) are a class of factors that regulate neuronal development, differentiation and function. NTs may activate two types of receptors, the high-affinity tyrosine kinase family TRK receptors and the low-affinity p75 neurotrophin receptor (p75NTR) of the tumor necrosis factor receptor superfamily (*Yang et al., 2016*). NTs can initiate various complex signal transduction pathways by activating these types of receptors and thus exert biological effects. In most cases, p75NTR is a ligand-activated apoptotic receptor, which primarily induces neuronal apoptosis and activate the apoptotic JKN-p53-Bax signal transduction pathway (*Redden et al., 2014*). TRKs mainly activate two pathways: the phosphoinositide 3-kinase (PI3K)-Akt signaling which inhibits the production and activity of apoptotic proteins, and the mitogen-activated protein kinase (MAPK) signaling pathway, which activates the anti-apoptotic proteins to promote survival (*Wong et al., 1999*). In this study, the KEGG pathway enrichment analysis of DEGs revealed that DEGs were the most enriched genes in the MAPK and PI3K-Akt

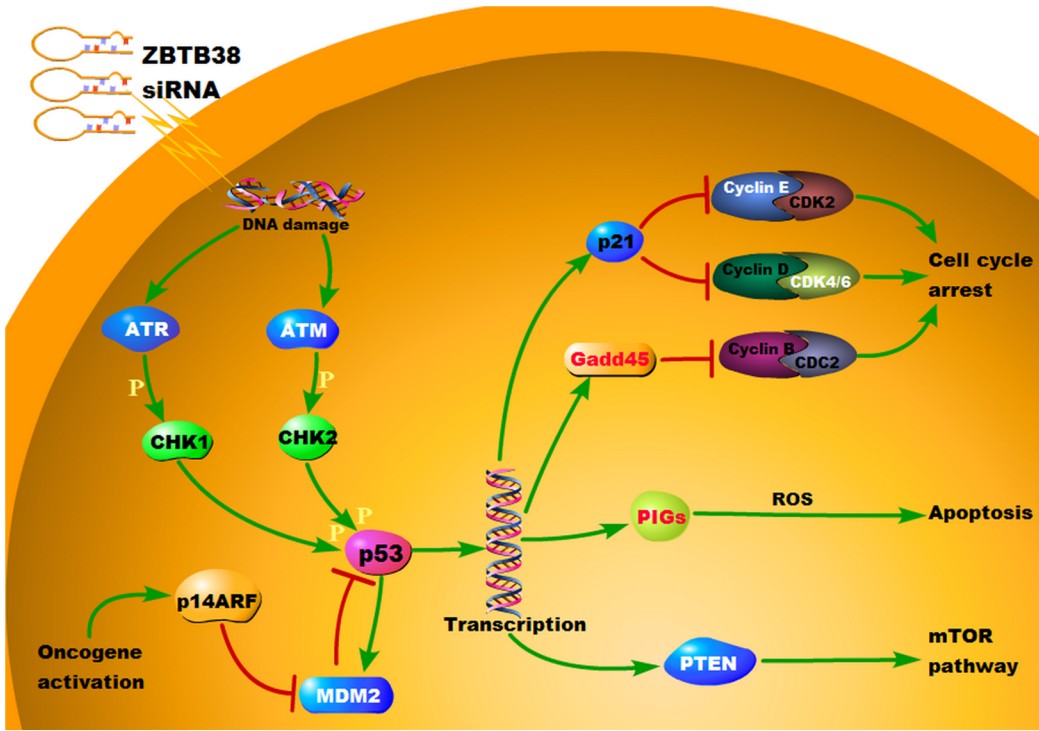

**Figure 10 KEGG pathway annotation map of differentially expressed genes in p53 signaling pathway.** Relative to the control group, the red labeled protein was associated with the up-regulated gene and the white labeled protein was associated with the down-regulated gene.

signaling pathways and mostly down-regulated. Thus, we speculated that ZBTB38 knockdown-induced reduction in viability and proliferation rate of SH-SY5Y cells may be closely related to these pathways. We plan to focus on the key components of DEGs in future studies to clarify the related molecular mechanism and to further evaluate the potential of ZBTB38 as a target gene to treating NB.

A key feature of NB is that it is a uniform p53 wild-type at diagnosis with intact intrinsic and extrinsic apoptotic mechanisms; direct inactivation of p53 mutations, which are rare, regardless of the stage of treatment, suggests that NBs feature an innate requirement for a baseline p53 activity (*Kim & Shohet, 2009*). In the present study, the KEGG pathway enrichment analysis of DEGs revealed that in the p53 signaling pathway, genes, including CDK4/6, Cyclin E, MDM2, ATM, ATR, PTEN, were down-regulated, and Gadd45 and PIGs were up-regulated after the knockdown of ZBTB38 (Fig. 10). Both the CDK4/6-Cyclin D and the CDK2-Cyclin E complexes serve as the central links in cell cycle regulation via regulating the G1–S transitions in cells, and abnormal activation of the CyclinD-CDK4/6–INK4–Rb pathway, which is often observed in various malignancies, will lead to uncontrolled growth of cancer cells (*Sawai et al., 2012*; *The et al., 2015*; *VanArsdale et al., 2015*). In addition, members of the Gadd45 family serve as key regulatory genes in DNA damage repair pathway with p53 as the central link, whereas the upregulation of Gadd45 plays an important role in the regulation of G2/M cell cycle checkpoints and the maintenance of genomic stability to inhibit the cell transformation

and the malignant tumor progression (*Wang et al., 1999*). ATM and ATR belong to the inositol trisphosphate kinase family, both of which can be activated by DNA damage to phosphorylate the downstream substrates such as CHK1, CHK2, and p53. In addition, the down-regulation of both kinases may impair the downstream transmission of the molecular signals and inhibit the p53 activity (*Abraham, 2001*; *Matsuoka et al., 2007*). MDM2 regulates the function of p53 via two approaches, that is, mediating the p53 degradation and inhibiting its transcriptional activity. As a negative feedback regulator of p53, the inhibited expression of MDM2 can enhance the transcriptional activity of p53 and inhibit tumorigenesis (*Shangary & Wang, 2009*). PIGs act as the target downstream genes of p53 for apoptosis regulation, which is critical for cell apoptosis by participating in the synthesis of reactive oxygen species (ROS) and the regulation of oxidative stress (*Jin et al., 2017*; *Lee et al., 2010*). We can speculate that when ZBTB38 gene was knocked down, more PIGs are transferred into the nucleus, where cell damage is repaired. Therefore, cellular response to DNA damage increased, and p53 induced ROS production, ultimately promoting the apoptosis of tumor cells. PTEN is a tumor suppressor gene with phosphatase activity. This gene is also an upstream regulatory inhibitor of the PI3K-Akt signal transduction pathway. PTEN is often referred to as a "switch" molecule in the PI3K-Akt pathway due to its capability, depending on its lipid phosphatase activity, to remove the phosphate group and participate in the regulation of cell activity. Once the expression of PTEN protein is reduced, the dephosphorylation of phosphatidylinositol (3,4,5)-trisphosphate (PIP3) decreases. Excessive PIP3 is subsequently accumulates in the cells, and the PI3K/Akt signaling pathway is continuously activated, eventually leading to cell proliferation, or uncontrolled apoptosis and finally the occurrence of various diseases (*Bleau et al., 2009*; *Carnero et al., 2008*). In summary, studies investigating the roles of ZBTB38 and p53 pathways in growth and apoptosis of NB cells and those involving the intervention of specific signaling pathways may allow us to further understand the mechanisms of NB occurrence and progression, and thus better evaluate and control this paediatric malignancy.

Among all the KEGG pathway enrichment categories, the majority of the DEGs were enriched in the PI3K-Akt signaling pathway, especially the down-regulated ones, with the most significance noted in PIK3C2A and RB1CC1. PIK3C2A is a member of the PI3K family and one of the key molecules in the signal transduction pathway of growth factors. The overexpression of PIK3C2A in cells has been reported to induce the accumulation and assembly of clathrin, which mediates the transport of proteins between cell membranes and the network structure of the Golgi body via regulating the movement of microtubules (*Dragoi & Agaisse, 2015*; *Shi et al., 2016*). RB1CC1 (also known as FIP200), with a molecular weight of 200kD, is an interacting protein of the focal adhesion kinase family. As documented in prior studies, autophagy induction is abolished in RB1CC1-deficient cells. RB1CC1 is an important regulatory protein that can acts on the autophagic initiation complex along with the Unc-51 like autophagy activating kinase simultaneously. RB1CC1 is also a key autophagy initiation factor in the mTORC1-dependent signaling pathway (*Ganley et al., 2009*; *Wang et al., 2011*; *Wei et al., 2009*). We also observed downregulation of PTEN, RAPTOR, and LC3B expression

(Fig. 9). RAPTOR is a specific component of mTORC1, which is negatively correlated with the activation of mTORC1. The downregulation of RAPTOR expression may also indicate the activation of mTORC1 pathway (Saxton & Sabatini, 2017). Therefore, as revealed in our study, we believe that the loss of ZBTB38 gene in SH-SY5Y cells lead to mTORC1-mediated autophagy inhibition.

Orthologous assignments of gene products were carried out using the COG database. Corresponding statistical analysis of the results also indicated that the silencing of the ZBTB38 gene affected the homeostasis of the whole cell. As a transcriptional factor, ZBTB38 regulates the transcription of intracellular proteins and influenced the expression and transport of proteins in the downstream signaling pathways. The GO functional enrichment analyses suggested that most of the DEGs were involved in "Binding" and "Catalytic Activity" of the MF between ZBTB38$^{-/-}$ cells and the controls. This finding also partially explains the biological functions of the key candidate genes enriched in the KEGG pathway, that is, all of them are specific binding DNAs or proteins that regulate, the transcriptional activity of target genes and are involved in various intracellular signaling pathways.

## CONCLUSIONS

The functional knockdown of transcription factor ZBTB38 effectively inhibited the proliferation and differentiation of NB cells, which may be largely attributed to the significant inhibition of the neurotrophin TRK receptor signaling pathway. In addition, the downregulation of ZBTB38 may also promote apoptosis of the NB cells by regulating key components of the p53 signaling pathway. Two DEGs (PIK3C2A and RB1CC1) that closely related to autophagy initiation were significantly inhibited, suggesting that ZBTB38 downregulation also blocked autophagy, an important mechanism that protects the cells from programmed cell death, thus accelerating apoptosis of tumor cells.

## ACKNOWLEDGEMENTS

This work was supported by the innovation team of the Scientific Research Platform in Anhui Province. The authors thank current and past members of Cai lab.

### Funding

This work was supported by the National Key R&D Program of China (No. 2018YFC1200201), National Natural Science Foundation of China under Grant (Nos. NSFC31372207 and 81570094) and a start-up grant from Nanjing Agricultural University (No. 804090). The funders had no role in study design, data collection and analysis, decision to publish, or preparation of the manuscript.

### Grant Disclosures

The following grant information was disclosed by the authors:
National Key R&D Program of China: 2018YFC1200201.

National Natural Science Foundation of China: NSFC31372207 and 81570094.
Nanjing Agricultural University: 804090.

## Competing Interests

The authors declare that they have no competing interests.

## Author Contributions

- Jie Chen conceived and designed the experiments, performed the experiments, analyzed the data, prepared figures and/or tables, authored or reviewed drafts of the paper, approved the final draft.
- Chaofeng Xing performed the experiments, analyzed the data, approved the final draft.
- Li Yan contributed reagents/materials/analysis tools, approved the final draft.
- Yabing Wang contributed reagents/materials/analysis tools, approved the final draft.
- Haosen Wang contributed reagents/materials/analysis tools, approved the final draft.
- Zongmeng Zhang performed the experiments, approved the final draft.
- Daolun Yu performed the experiments, approved the final draft.
- Jie Li performed the experiments, approved the final draft.
- Honglin Li contributed reagents/materials/analysis tools, approved the final draft.
- Jun Li contributed reagents/materials/analysis tools, approved the final draft.
- Yafei Cai conceived and designed the experiments, authored or reviewed drafts of the paper, approved the final draft.

## Data Availability

The clean data of this article are publicly available in the NCBI Sequence Reads Archive (SRA) with accession number: SRP150042. Supplemental Material has been provided with the article and is also available at Figshare:

Chen, Jie; Xing, Chaofeng; Yan, Li; Wang, Yabing; Wang, Haosen; Zhang, Zengmeng; et al. (2018): Supplemental Material for Jie Chen et al., 2018. figshare. Fileset. DOI 10.6084/m9.figshare.6849089.v1

## Supplemental Information

Supplemental information for this article can be found online at http://dx.doi.org/10.7717/peerj.6352#supplemental-information.

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
