# Peer review of "Transcriptome profiling reveals the role of ZBTB38 knock-down in human neuroblastoma"

_PeerJ, doi:10.7717/peerj.6352_

## Round 0.1 · original submission · Major Revisions

The manuscript has been carefully evaluated by two external reviewers. The reviewers found value in your work but substantial major revisions are requested. A more detailed experimental design needs to be implemented to convince about the conclusions on apoptosis activation, autophagy inhibition and p53 with key roles in this mechanism. Reviewer #1, in particular, indicated a clear plan for the experiments that can be included, which are important to address point by point. Moreover, the authors before resubmitting a revision should have help from a professional scientific English proofreader.

Reviewer 1 ·

Basic reporting

Authors found that ZBTB38 gene is downregulated in different tumors and its low expression is associated with good prognosis in LGG (Brain lower grade glioma). They studied the ZBTB38 downregulation effects on SH-SY5Y cells: ZBTB38 SiRNA led to a decrease of cell proliferation and viability. Furthermore, the RNA-sequencing has deepened the analysis of ZBTB38 silencing effects, offering possible molecular mechanisms that could be implicated in the signaling cascade.
This story would be of interest, however, there are some points that need clarification and additional experiments to strengthen the claims.
Authors should better described the proposed experiments:
e.g., They have “a total of 2,036 (83.5%) down-regulated DEGs and 402 (16.5%) up-regulated DEGs were selected”. GO analysis is confusing; it is not clear on which group they have done the analysis so it is difficult to understand the data.

Experimental design

Major issues:
- ZBTB38 is remarkable downregulated in UCEC and CESC tumor, but authors write “low expression of ZBTB38 was associated with improved the prognosis of the LGG (Brain lower grade glioma) patients (Fig. 2), suggesting that these changes are closely related to neuronal tumors.” What about UCEC and CESC? How do authors explain this point?
- Taking into account the results, at present data do not support the working model in Figure 8: p53 pathway is proposed to be a key signaling, but it has not been identified by GO enrichment and KEGG pathway enrichment analysis, why?
- How autophagy is related to ZBTB38? Authors should check for autophagic flux by WB or ICC to verify that autophagy is impaired upon ZBTB38 downregulation. In S3 Table. (Top 20 differentially expressed genes associated with autophagy), there are genes that can have opposite effects on autophagy: for example, RB1CC1 (FIP200), ATG2 and ATM positively regulate autophagy induction whereas PIK3CA and PIK3C2A mediate the activation of AKT-mTORC1 cascade that inhibits autophagy through ULK1 (Laplante et al., 2012). Moreover, lose of PTEN, a phosphates that inhibits growth and proliferation signals of phosphoinositide 3‑kinase (PI3K) by dephosphorylating phosphatidylinositol 3,4,5‑trisphosphate (Crino, 2016), should promote the AKT-MTORC1 activation. Thus, this point is unclear and needs further investigations to be clarified; I suggest checking AKT-mTORC1 activation state by WB.
- They should provide results for ROS increase and apoptosis activation upon ZBTB38 silencing.
- Figure legends are not complete or are not clear enough to read.
Minor points:
There are some typing errors.

1 Laplante M, Sabatini DM. mTOR signaling in growth control and disease. Cell 2012; 149: 274–93.
2 Crino PB. The mTOR signalling cascade: Paving new roads to cure neurological disease. Nat Rev Neurol 2016; 12: 379–392.

Validity of the findings

At this point, the results are not enough robust to support the conclusions.

Reviewer 2 ·

Basic reporting

The manuscript is well prepared, and data presentation is pretty clear.

Experimental design

The study is well organized, and experiment design is reasonable.

Validity of the findings

Data is robust, statistically sound, & controlled.

Additional comments

In this paper the authors aim to demonstrate the possible role of a zinc finger protein ZBTB38 in neuroblastoma (NB). Based on the human neuroblastoma cell line SH-SY5Y model, high throughput RNA sequencing was performed to compare the differential gene expression between the control and ZBTB38 knockdown cell lines. The data revealed a major transcriptional activation role of ZBTB38, with 83.5% down-regulated out of the 2,438 identified differentially expressed genes. Functional annotation revealed the involvement of the neurotrophin TRK receptor related signaling pathway, including PI3K/Akt and MAPK signaling pathway, several other well-known genes like PTEN and p53, and autophagy-related genes PIK3C2A and RB1CC1, providing a pilot investigation about the function role of ZBTB38 in neuroblastoma development and clue for targeted therapy. The study is well organized, data presentation is pretty clear, and results are very interesting.
1. The NGS sequencing data should be submitted and deposited to public data base for easy accessibility;
2. For Figure 1B-E, a P value should be calculated and provided to support their claim of differential expression;
3. For functional annotation of the differentially expressed gene, the author may want to try GSEA analysis module, which will generate a more informative enrichment plot with enrich score and p value, and a heatmap for the enriched gene list, compared to the relative simple bar graph of the genes in each pathway presented here.

---

## Round 0.2 · accepted · Accept

I am glad to accept the manuscript, which has been improved upon revision.

# Reviewer 2 ·

Basic reporting

no comment

Experimental design

no comment

Validity of the findings

no comment

Additional comments

This manuscript is greatly improved and meets the standard to be published in this journal. The very informative data is valuable to the neuroblastoma field.